

# Impact of the antidepressant citalopram on the behaviour of two different life stages of brown trout

Michael Ziegler[1], Sarah Knoll[2], Heinz-R. Köhler[1], Selina Tisler[3], Carolin Huhn[2], Christian Zwiener[3] and Rita Triebskorn[1,4]

[1] Animal Physiolgical Ecology, Eberhard-Karls-Universität Tübingen, Tübingen, Baden-Württemberg, Germany
[2] Effect-based Environmental Analysis, Eberhard-Karls-Universität Tübingen, Tübingen, Baden-Württemberg, Germany
[3] Environmental Analytical Chemistry, Eberhard-Karls-Universität Tübingen, Tübingen, Baden-Württemberg, Germany
[4] Steinbeis Transfer Center for Ecotoxicology and Ecophysiology, Rottenburg, Baden-Württemberg, Germany

Corresponding author
Michael Ziegler,
michael.ziegler@student.uni-tuebignen.de

## ABSTRACT

**Background**. Over the last two decades, there has been a constant increase in prescription rates of antidepressants. In parallel, neuroactive pharmaceuticals are making their way into aquatic environments at increasing concentrations. Among the antidepressants detected in the environment citalopram, a selective serotonin reuptake inhibitor, is one of the most commonly found. Given citalopram is specifically designed to alter mood and behaviour in humans, there is growing concern it can adversely affect the behaviour on non-target wildlife

**Methods**. In our study, brown trout were exposed to citalopram (nominal concentrations: 1, 10, 100, 1000 µg/L) in two different life stages. Larvae were exposed at 7 and 11 °C from the eyed ova stage until 8 weeks post yolk sac consumption, and juvenile brown trout were exposed for 4 weeks at 7 °C. At both stages we measured mortality, weight, length, tissue citalopram concentration, behaviour during exposure and behaviour in a stressfull environment. For brown trout larvae additionally hatching rate and heart rate, and for juvenile brown trout the tissue cortisol concentration were assessed.

**Results**. During the exposure, both larvae and juvenile fish exposed to the highest test concentration of citalopram (1 mg/L) had higher swimming activity and spent longer in the upper part of the aquaria compared to control fish, which is an indicator for decreased anxiety. Most probably due to the higher swimming activity during the exposure, the juveniles and larvae exposed to 1 mg/L citalopram showed decreased weight and length. Additionally, in a stressful artificial swimming measurement device, brown trout larvae displayed the anxiolytic effect of the antidepressant by reduced swimming activity during this stress situation, already at concentrations of 100 µg/L citalopram. Chemical analysis of the tissue revealed rising citalopram tissue concentrations with rising exposure concentrations. Tissue concentrations were 10 times higher in juvenile fish compared to brown trout larvae. Fish plasma concentrations were calculated, which exceeded human therapeutic levels for the highest exposure concentration, matching the behavioural results. Developmental parameters like hatching rate and heart rate, as well as mortality and tissue cortisol content were unaffected by the antidepressant.

Overall, we could trace the pharmacological mode of action of the antidepressant citalopram in the non-target organism brown trout in two different life stages.

## INTRODUCTION

Pharmaceuticals like psychotropic drugs are widely distributed in the environment and can be found in most human-influenced surface waters and within most trophic levels, from algae to fish (*Alvarez-Munoz et al., 2015*; *aus der Beek, Grüttner & Carius, 2016*). During the last decades, psychotropic drugs, especially antidepressants, increasingly detected in rivers and surface waters (*Acuna et al., 2015*; *Grabicova et al., 2017*; *Hughes, Kay & Brown, 2013*; *Schultz et al., 2010*) due to the increasing numbers of diagnosed mental disorders (*Destatis, 2017*) and the consequent rising numbers of antidepressant prescriptions (*Schwabe & Paffrath, 2016*). Within the group of antidepressant drugs, the selective serotonin reuptake inhibitors (SSRIs) make up the largest part of the prescription rates (*Schwabe & Paffrath, 2016*). Their mode of action is based on their binding to the serotonin transporter (SERT) in the presynaptic membrane, thereby inhibiting the reuptake of serotonin into the presynaptic neuron causing increased serotonin levels in the synaptic cleft (*Hyttel, 1994*). These 5-HT reuptake transporters are highly conserved in the animal kingdom and can be found in all phyla (*Gunnarsson et al., 2008*; *Verbruggen et al., 2018*). One of the most important antidepressants of the SSRI class is citalopram, which is the most prescribed antidepressant in Germany, with 306.8 million defined daily doses (DDD) in 2015 (*Schwabe & Paffrath, 2016*). Assuming a DDD of 20 mg per patient, this leads to a minimum consumption of 6.1 t/year in Germany alone.

Citalopram has been found in US surface waters in concentrations ranging from 4 ng/L to 219 ng/L (*Schultz et al., 2010*). In different effluents of wastewater treatment plants, citalopram was found in concentrations ranging from 44 ng/L to 431 ng/L (*Himmelsbach, Buchberger & Klampfl, 2006*; *Nodler et al., 2010*; *Silva et al., 2014*). Furthermore, surface water citalopram was detected at concentrations of up to 76 µg/L downstream of a wastewater treatment plant in India, close to Hyderabad and up to 840 µg/L in the effluent of a drug manufacturer (*Fick et al., 2009*; *Larsson, De Pedro & Paxeus, 2007*). In contrast environmental concentrations of other antidepressants like fluoxetine are lower within a range of 1 to 43 ng/L (*Acuna et al., 2015*; *Paiga et al., 2016*; *Schultz et al., 2010*). With regard to the critical effect concentrations (CEC) published by *Fick et al. (2010)*, it can be seen that citalopram has a very low CEC of 141 ng/L compared to other antidepressants like fluoxetine (CEC: 489 ng/L) or venlafaxine (CEC: 6112 ng/L). Despite the fact that citalopram is one of the most commonly prescribed antidepressants and is frequently detected in surface waters, most studies on the effects of SSRIs in fish were performed with other pharmaceuticals like fluoxetine (*Airhart et al., 2007*; *Foran et al., 2004*; *Henry*

& *Black, 2008*; *Martin et al., 2017*); as a result, there is a deficit in knowledge concerning the effects of citalopram in fish. At present, the impact of citalopram on fish have been mixed: *Kellner et al. (2016)* showed reduction in anxiety related behaviour indicated by an increase in the swimming activity in three-spined sticklebacks and a longer time spent in the upper part of the aquaria, caused by exposure to 1.5 µg/L citalopram for 21 days. In addition, anxiolytic effects like reduced freezing behaviour in the novel tank diving test and increased curiosity in the novel object test were observed after exposure to 15 and 1.5 µg/L of citalopram respectively for 21 days. Whereas, *Kellner et al. (2017)* exposed three-spined sticklebacks in a developmental stage for 30 days to 1.5 µg/L citalopram with a subsequent 120 days recovery phase and showed a reduced swimming activity in the novel tank diving test as well as increased aggressive behaviour. Though, neither freezing behaviour and latency or number of transitions spent in the upper half were influenced, which can be seen as no change in anxiety. But, *Olsen et al. (2014)* could show anxiolytic effects in the novel tank diving test, like reduced freezing behaviour and faster and longer time spent in the upper aquaria in Endler's guppies exposed to 2.3 and 15 µg/L of citalopram for 21 days. With regard to feeding behaviour, *Kellner et al. (2015)* showed that three-spined sticklebacks had reduced food intake when exposed to 0.15 µg/L citalopram for 21 days. But in contrast, *Kellner et al. (2017)* observed increased food intake in sticklebacks exposed to 1.5 µg/L citalopram for 30 days during developmental stages, and a subsequent 120 days recovery phase. *Keysomi, Sudagar & Asl (2013)* also proved a decrease in plasma cortisol level in rainbow trout (*Oncorhynchus mykiss*) exposed to 5 µg/L citalopram for 10 days. Due to these diverse outcomes, it is important to assess different endpoints in different life stages to detect the diverse effects that citalopram can have on aquatic organisms. In our case, two life stages of the native fish species brown trout (*Salmo trutta* f. *fario*) were chosen because of the species' sensitivity and high ecological relevance in central Europe (*Klemetsen et al., 2003*). To assess the effects of citalopram on developmental parameters like mortality, time to hatch and heart rate, we exposed brown trout eggs to citalopram at 7 and 11 °C for 5 months until 8 weeks post yolk consumption. Also, juvenile brown trout were exposed to citalopram for 4 weeks at 7 °C, and the apical endpoints mortality, weight and length were evaluated. The cortisol content of the juvenile fish was assessed. Cortisol is a glucocorticoid hormone which has various effects in fish like the regulation of hydro-mineral balance and energy metabolism (*Wendelaar Bonga, 1997*), and it can be used as an indicator of stress. In addition, in both experiments behavioural parameters were recorded during the exposure period. The preference of the fish to stay in the upper aquaria part in the exposure tanks was assessed and swimming behaviour in a stressful environment was recorded using an artificial swimming measurement device. The aim of this study was to evaluate effects of citalopram on developmental and behavioural endpoints. Furthermore, the study aimed at showing whether these effects occur under chronic exposure at environmentally relevant concentrations.

## MATERIAL AND METHODS

### Fish

Brown trout (*Salmo trutta* f. *fario*) eggs and juveniles were purchased from a trout farm in Southern Germany (Forellenzucht Lohmühle, Alpirsbach-Ehlenbogen, Germany). This commercial fish breeder is listed as category I (disease-free) according to the EC Council Directive (2006). The eggs were obtained in the eyed ova stage in December 2016 and directly transferred into the experiment. The juveniles were acclimatised to laboratory conditions for two weeks prior to exposure in a 200 L tank (filtered tap water, aerated) and exposed afterwards in August 2017. Fish were kept under a 10:14 light:dark regime and fed daily with commercial trout feed (0.8 mm, Inico Plus, Biomar, Brande, Denmark). All animals were approved by the animal welfare committee of the Regional Council of Tübingen, Germany (ZO 2/16).

### Test Substance

Citalopram hydrobromide ($C_{20}H_{21}FN_2O \cdot HBr$, CAS: 59729-32-7) was purchased from Sigma Aldrich (Steinheim, Germany). It was dissolved in distilled water to obtain stock solutions of 100 mg/L and 1 mg/L citalopram. The citalopram concentrations refer to citalopram free base ($C_{20}H_{21}FN_2O$). To achieve the respective nominal concentrations, test solutions were prepared with appropriate volumes of the equivalent stock solutions and aerated, filtered tap water (iron filter, active charcoal filter, particle filter).

### Experiment with brown trout larvae

Brown trout eggs in the eyed ova stage (37 days post fertilisation (dpf)) were exposed in a semi static setup with three replicate each of 0, 1, 10, 100, 1,000 µg/L citalopram at both 7 °C or 11 °C, in order to reveal influences of temperature on the effects investigated. Additionally, we had one tank in each of the three replicate blocks with 100 µg/L citalopram but without fish, which served as a control for ingestion, photolytic and microbial degradation of the chemical. Aquaria containing 10 L test solution and 30 fish each were set up in triplicate in a randomised order. Twice a week, 50% of the test solution were replaced with freshly prepared test solution. A 10:14 light:dark cycle was set and the tanks were covered with black foil to protect them from direct light. Fish were fed daily (0.5 mm, 0.8 mm, Inico Plus, Biomar, Brande, Denmark) from the day the yolk-sac was consumed (for 7 °C: 52 d post hatch; for 11 °C: 35 d post hatch) with a defined amount of food (3% body weight) adjusted to the developmental state of the fish. Exposure ended 8 weeks (total exposure time 7 °C: 135 d; 11 °C: 107 d) after yolk-sac consumption. During the exposure, time to hatch and mortality were recorded daily. At 7 days post-hatch, the heart rate of 5 individuals of each control and the highest concentration tank was measured and, whenever a difference was revealed, the other treatments were also assessed. Two weeks before sampling, pictures of the photographable tanks were taken on a daily basis to assess the number of fish in the upper and lower aquaria part during exposure. One week before the last sampling, the swimming behaviour was recorded using an artificial swimming measurement device (ASMD). Here, 5 fish from each tank were transferred into small glass aquaria and swimming behaviour was recorded for 18 min. Terminally,

fish were sampled 8 weeks after yolk sac consumption. When sampling took place, the fish were anaesthetised by an overdose of the fish anaesthetic MS222 (tricaine mesylate, 1 g/L, buffered with $NaHCO_3$) followed by a cervical spine cut. After individual determination of the weight and the total length fish were dissected, and tailfins were frozen in liquid nitrogen and stored at $-80\,°C$ for further analysis of citalopram tissue content. Water conditions (temperature, conductivity, pH, oxygen content) were tested at the beginning, twice during the experiment and at the end.

## Experiment with juvenile brown trout

Juvenile brown trout (8 months post hatch) were exposed at $7\,°C$ in a semi-static three block setup to 0, 1, 10, 100, 1,000 µg/L citalopram for 28 days. The treatments were setup in triplicate in a randomised order in aquaria containing 15 L of the test solution and 10 fish each. Twice a week, 50% of the test solution were renewed. The test was conducted under a 10:14 light:dark regime at $7\,°C$, and the tanks were covered with black foil to protect them from direct light. Fish were fed daily with a defined amount (3% body weight) of commercial trout feed (0.8 mm, Inico Plus, Biomar, Brande, Denmark). Mortality was recorded daily. From two weeks before the sampling took place until the end of the experiment, daily pictures of the photographable tanks were taken to assess the number of fish in the upper and lower aquaria portion during the exposure. In addition, three fish from each tank were used for swimming behaviour measurements in the artificial swimming measurement device (ASMD) explained further below and sampled afterwards. At the end of the experiment, 7 of the 10 fish per tank were anaesthetised and killed by an overdose of the fish anaesthetic, MS222 (Tricaine mesylate, 1 g/L, buffered with $NaHCO_3$) followed by a cervical spine cut. Prior to dissection, the weight and total length of fish were determined. The dorsal part and tailfin of the fish were frozen in liquid nitrogen and stored at $-80\,°C$ for further analysis of cortisol and citalopram tissue content. Water conditions were tested at the beginning and end of the experiment, and water samples for chemical analyses were taken right before the start of the experiment, after 2 weeks and at the end of the experiment.

## Chemical analyses

Water samples were taken at the beginning, in the course and at the end of the experiments. Sampling during the exposure period took place regularly before and after water exchange. Water samples from triplicate aquaria were pooled and stored at $-20\,°C$ until further processing. For tissue analysis, at the end of the experiments, tailfin samples of the fish were taken to determine the citalopram concentration in the muscle.

## Water analysis

The real water concentrations were determined using LC-MS with a 1290 Infinity HPLC system (Agilent Technologies, Waldbronn, Germany) and a triple quadrupole mass spectrometer (6490 iFunnel Triple Quadrupole LC/MS, Agilent Technologies, Santa Clara, CA, USA) in ESI (+) mode. An Agilent Poroshell-120-EC-C18 column (2. 1× 100 mm; 2.7 µm particle size) was used at a flow rate of 0.4 mL/min for separation, and column temperature was maintained at $40\,°C$. Eluent A and B were water (+0.1% formic acid) and

acetonitrile (+0.1% formic acid), respectively. Gradient elution was used: 0–1 min 5% B, linear increase to 100% B within 7 min, hold for 7 min at 100% B. After switching back to the starting conditions, a reconditioning time of 3 min was employed. Samples were kept in the autosampler at 10 °C. The injection volume was 1 or 10 μL (dilution factor 0–100). The limit of detection of citalopram (mass transition m/z 325 → 109) for undiluted samples was 10 ng/L (10 μL injection volume). Further details on the operating parameters of the triple quadrupole are provided in the supplement.

## Tissue analysis

The citalopram concentrations in the tissues of brown trout larvae and juveniles were determined by liquid chromatography–mass spectrometry (LC–MS). For sample extraction, a miniaturised and optimised QuEChERS procedure was applied. Fish samples (tailfin samples containing mainly muscle tissue) originating from all exposure concentrations were analysed. For each exposure group, tissue samples of 10 individuals per treatment were pooled. Frozen fish samples (−20 °C) were first homogenised by grinding using a mortar and pestle under liquid nitrogen. Aliquots of the homogenised samples were transferred to an Eppendorf tube, and 0.25 mL acetonitrile and 0.75 mL water were added. For extraction, samples were shaken with a vortex device for 30 sec., after which 30 mg sodium chloride and 120 mg anhydrous magnesium sulfate were added; and the sample was immediately shaken for 30 s. After centrifugation for 15 min at 13,000 rpm, 0.1 mL of the acetonitrile phase were evaporated to dryness under a gentle stream of nitrogen and the concentrated residue was resolved in 0.3 mL methanol. The extracts were diluted to reach concentrations compatible with the calibration range established for citalopram, and filtered for LC-MS analysis. Matrix matched calibration was performed between 1 and 20 μg/L. The limit of detection was 0.06 ng/g. Further details can be found in the supplementary material.

All analyses were performed using a 1260 Infinity LC system coupled to a 6550 iFunnel QTOF mass spectrometer (Agilent Technologies, Waldbronn, Germany and Santa Clara, CA, USA) with an electrospray ionisation source (ESI). Aliquots of 10 μL sample were injected onto a Zorbax Eclipse Plus C18 column (2.1 × 150 mm; 3.5 μm particle size, narrow bore, Agilent Technologies, Waldbronn, Germany) at a column temperature of 40 °C. A gradient elution at a flow rate of 0.3 mL/min using water and methanol, containing 0.1% formic acid, was used. Details on the LC-MS method are given in the supplementary material.

## Swimming behaviour in exposure aquaria

For quantification of the swimming behaviour during the exposure, photos of tanks were taken and the number of fish in the upper and lower aquaria part was counted. This was only possible for some aquaria (8 of 15 for brown trout larvae, 9 of 15 for juvenile brown trout) due to their position in the climate chamber. Nevertheless, the selection of photographable tanks was representative for the entire number of aquaria, because for control and the highest treatment at least 2 tanks were photographed. Pictures were taken with a Panasonic DMC-TZ56 camera 5 min after the black foil cover was removed; a white

sheet of paper was placed in the back of the aquarium to provide a bright background for better contrast. For experiments with brown trout larvae, 3 pictures were taken per day of each photographable tank at an interval of 5 min, from one-week prior to the experiment until sampling (Apr 03–Apr 13, 2017). In the juvenile brown trout experiment, 3 pictures were taken every day of each photographable tank at an interval of 5 min, from two weeks before sampling until sampling (Aug 21–Sep 03, 2017). The pictures were analysed manually and the number of fish located in the lower and upper half of the aquaria was recorded. Data for all pictures taken from one tank on the same day were averaged.

## Artificial swimming measurement device (ASMD)

The recording of the brown trout larvae took place one week before the second sampling of the fish. Small aquaria (17 × 17 × 8.5 cm) were filled with 500 mL of the respective test solution at an appropriate temperature, and five brown trout larvae were placed in there. The testing of juvenile fish was scheduled after the sampling, where the swimming behaviour of the three leftover juvenile brown trout from each tank was recorded. One litre of the respective test solution was added to the small aquaria, before the three juvenile fish were transferred to them and recorded. Each of the four small aquaria was equipped with a camera (Basler acA 1300–60 gm, 1.3 MP resolution, Basler AG, Ahrensburg, Germany, lens: 4.5–12.5 mm; 1:1.2; IR 1/2″) placed 32 cm above the water surface. The set-up was arranged on a table in the climate chamber and enclosed by white polystyrene plates on each side and on top. Inside the enclosure, 4 lamps (2700 K, 1521 lm each) were placed, one in each corner facing the top polystyrene plate to obtain indirect illumination. The bright illumination, lack of aeration of the ASMD-aquaria and the transfer process of the fish led to stressful conditions for them. Locomotion was recorded for 20 min, but the first 2 min were ignored to account for acclimatisation. During the remaining 18 min, video sequences were taken and each of the four aquaria was analysed individually. Fish were centre-point tracked individually, and the total distance moved, the average velocity and the time of no movement were logged with the EthoVision 12 XT (Noldus Information Technology bv, Wageningen, Netherlands). A manual correction of some of the tracked data was essential due to difficulties in automatic tracking.

## Cortisol content

Cortisol content was determined in juvenile brown trout exposed under stressful conditions in the ASMD as well as in fish not exposed to such stress. The cortisol content was measured with the commercially available Fish cortisol ELISA Kit by Cusabio Technology LCC (Houston, Texas, USA). The dorsal parts (muscle and kidney tissue) of juvenile brown trout were manually homogenised in 1xPBS buffer (tissue/buffer ratio 1:11 w/v) with a pestle. After 2 freeze-thaw cycles at $-20\,°C$ and room temperature, the samples were centrifuged (5,000×g, 5 min, 4 °C) and the supernatant stored at $-20\,°C$ until analysis. Before pipetting the assay, the supernatant was diluted with sample buffer provided in the Kit (supernatant/buffer ratio 1:10 v/v). The assay was conducted in a pre-coated 96 well plate provided by the manufacturer. Each well contained 50 μL antibody and either 50 μL of standard or 50 μL of sample, before being incubated for 40 min at 37 °C.

After 3 washing cycles with washing buffer, 100 µL of HRP-conjugate was added and incubated for 30 min at 37 °C. Following 5 washing cycles with washing buffer, 90 µL of TMB (3,3′,5,5′-Tetramethyl[1,1′-biphenyl]-4,4′-diamine) substrate were added and incubated for 20 min at 37 °C. Then, 50 µL of stop solution were added to each well and the plate was measured photometrically at 450 nm and for wavelength correction at 570 nm. Concentrations were calculated with blanked and wavelength corrected data to a four parameter logistic standard curve fit. Concentrations of cortisol are expressed in ng/mL (see Table 1).

## Statistical analysis

Statistical analyses were performed with SAS JMP 14 and R 3.5.0 (packages: lme4). Mortality and time to hatch were analysed by nested Cox proportional hazards model, using replicate aquaria as a nested factor. Length, weight and total distance moved of the ASMD were analysed by a nested ANOVA, using replicate aquaria as nested factor, and a post hoc Dunnett's test. If necessary, data were transformed to achieve normal distribution and homogeneity of variance. If no normal distribution could be achieved, data were evaluated with a nonparametric Kruskal-Wallis test post-hoc Steel method with control. The difference in cortisol content, mean velocity and no movement ofer time of the ASMD was analysed with a Linear Mixed Model with replicate as random factor and subsequently post-hoc Dunnett's test. Data for swimming behaviour during the exposure were evaluated with a Generalized Linear Mixed Model (binomial distribution, aquarium identity as random factor) and supsequently post-hoc Dunnett's Test. The $\alpha$-level was set to 0.05. Comparison of the results for different climate chambers was only descriptive to prevent the problem of pseudo-replication due to missing climate chamber replicates. Statistical details are given in the supplementary material.

## Criteria for reporting and evaluating ecotoxicity data (CRED)

Criteria for reporting and evaluating ecotoxicity data (CRED) are given in the Supplemental Information (*Moermond et al., 2016*). CRED is important to improve the reproducibility, relevance and transparency of aquatic ecotoxic research between the different institutions (*Moermond et al., 2016*).

# RESULTS

## Water conditions

Temperature, conductivity, pH and oxygen content were measured at the beginning and end of both experiments. In the brown trout larvae experiment, water quality parameters were assessed at 2 additional time points (18.01.2017, 06.03.2017). All water quality parameters were in an acceptable range (brown trout larvae: mean temperature 7 °C: 7.1 ± 0.32 °C; 11 °C: 10.47 ± 0,24 °C; mean conductivity 7 °C: 472.6 ± 9.9 µS/cm; 11 °C: 478.3 ± 7.2 µS/cm; mean pH 7 °C: 8.08 ± 0.41; 11 °C: 7.96 ± 0.46; mean oxygen content 7 °C: 10.77 ± 0.3 mg/L; 11 °C: 9.94 ± 0.5 mg/L; juvenile brown trout: mean temperature: 7.15 ± 0.41 °C; mean conductivity: 493.7 ± 17.5 µS/cm; mean pH: 8.09 ± 0.01; mean oxygen content: 11.22 ± 0.1 mg/L). Further details are given in the supplementary materials.

Ziegler et al. (2020), *PeerJ*, DOI 10.7717/peerj.8765
**Table 1  Results for brown trout larvae exposed to citalopram.**

| Temperature | 11 °C | | | | | 7 °C | | | | |
|---|---|---|---|---|---|---|---|---|---|---|
| Treatment (μg/L) | 0 | 1 | 10 | 100 | 1000 | 0 | 1 | 10 | 100 | 1000 |
| Mortality (%) | 26.67 ± 2.72 | 32.22 ± 1.57 | 28.89 ± 8.75 | 35.56 ± 4.16 | 22.22 ± 3.14 | 2.22 ± 3.14 | 4.44 ± 4.16 | 3.37 ± 2.72 | 6.74 ± 2.72 | 6.67 ± 2.72 |
| Weight (g) | 0.449 ± 0.139 | 0.484 ± 0.169 | 0.447 ± 0.140 | 0.487 ± 0.143 | **0.306 ± 0.146 ***** | 0.327 ± 0.089 | 0.413 ± 0.113 | 0.335 ± 0.101 | 0.337 ± 0.091 | **0.247 ± 0.066 ***** |
| Length (cm) | 3.76 ± 0.37 | 3.83 ± 0.42 | 3.73 ± 0.39 | 3.74 ± 0.40 | **3.24 ± 0.43 ***** | 3.33 ± 0.27 | 3.38 ± 0.32 | 3.30 ± 0.32 | 3.25 ± 0.28 | **2.91 ± 0.23 ***** |
| Heart rate (bpm) | 76 ± 3.74 | n.a. | n.a. | n.a. | 75 ± 4.52 | 49.93 ± 1.5 | 51.8 ± 3.19 | 49.2 ± 3.25 | 50 ± 3.03 | 49 ± 2.37 |
| Time to hatch (dpf) | 49.69 ± 0.93 | 49.64 ± 1.11 | 49.74 ± 1.02 | 49.22 ± 0.96 | 49.14 ± 0.84 | 59.38 ± 1.27 | 58.68 ± 1.65 | 59.07 ± 1.31 | 58.60 ± 1.31 | 59.38 ± 1.17 |
| Aquaria photographed | 2 | n.a. | 2 | 2 | 2 | 2 | 1 | 2 | 1 | 2 |
| Fish in upper aquaria half (%) | 7.29 ± 11.11 | n.a. | 5.71 ± 10.36 | 10.68 ± 13.24 | **66.52 ± 11.28 ***** | 0.14 ± 0.68 | **2.3 ± 3.73 *** | **2.65 ± 3.22 *** | 1.30 ± 2.47 | **79.36 ± 7.19 ***** |
| ASMD: total distance moved (cm) | 2700 ± 1405 | 1987 ± 755 | 2501 ± 815 | 2252 ± 1353 | **1133 ± 1015 ***** | 2473 ± 1016 | 1591 ± 1045 | 1792 ± 1320 | **1361 ± 1033 *** | **568 ± 697 ***** |
| Aqueous citalopram concentration (μg/L) | <LoD | 0.97 ± 0.20 | 8.30 ± 1.17 | 65.74 ± 5.77 | 973.98 ± 180.64 | <LoD | 0.83 ± 0.27 | 8.74 ± 0.48 | 70.50 ± 11.11 | 1017.97 ± 125.84 |
| Tissue citalopram concentration (μg/g) (wet weight) | <LoD | 0.07 ± 0.014 | 0.69 ± 0.1 | 1.57 ± 0.451 | 55.87 ± 12.972 | <LoD | 0.2 ± 0.042 | 0.97 ± 0.235 | 5.63 ± 2.0 | 142.15 ± 44.961 |

**Notes.**

Data are shown as arithmetical means ± standard deviation.

Asterisks represent significant differences to the respective control (*$p < 0.05$; **$p < 0.01$; ***$p < 0.001$).

Abbreviations: n.a., not assessed; dpf, day post-fertilisation; LoD, limit of detection.

## Chemical analyses

Regarding water analysis, citalopram could not be detected in any of the control samples. In most of the treatments, aqueous citalopram concentrations measured were lower than nominal concentrations, except for the treatments with the highest concentrations in the brown trout larvae experiment and the exposure at 1 μg/L of the juvenile brown trout experiment. The recovery rate was about 80%. The citalopram concentrations in the controls for photolytic and microbial degradation were slightly higher (79.84 ± 2.50 μg/L) than in the 100 μg/L exposure tank (70.50 ± 11.11 μg/L). Overall, the measured citalopram concentrations in water samples were in good accordance with the nominal concentrations (Table 1). Further details on water concentrations are given in the supplementary materials.

Regarding biota analysis, citalopram determined in tissue samples was in the μg/g range. Citalopram could not be detected in the muscle tissue of brown trout in any of the control samples. Tissue concentrations of citalopram were shown to correlate with water concentrations with the highest values in fish exposed to 1,000 μg/L citalopram. Tissue concentrations of brown trout larvae exposed at 7 °C were higher than those of brown trout larvae exposed at 11 °C. Juvenile brown trout accumulated at least 20 times more citalopram in muscle tissue than brown trout larvae.

## Experiment with brown trout larvae

The mortality of brown trout larvae was not affected by citalopram (11 °C Cox Regression: $df = 4$, $\chi 2 = 4.2743$, $p = 0.370$; 7 °C Cox Regression: $df = 4$, $\chi 2 = 6.9203$, $p = 0.140$). However, the mortality of larvae exposed at 11 °C was higher in all treatments, including the controls, from day 51 to day 86 of exposure (43 d–71 d post hatch) (11 °C mean mortality 29.11%). The mortality of fish exposed to 7 °C ranged from 0–10% (7 °C mean mortality 4.69%). Also, time to hatch did not differ between treatments (11 °C: Cox Regression: $df = 4$, $\chi 2 = 0$, $p = 1$; 7 °C: Cox Regression: $df = 4$, $\chi 2 = 2.42E-0.9$, $p = 1$). The fish exposed at 11 °C hatched approximately 10 days earlier than fish exposed at 7 °C. The heart rate of brown trout larvae was not affected when exposed to 1,000 μg/L citalopram at 11 °C (nested ANOVA: $df = 1$, $F = 0.3968$, $p = 0.535$) and 7 °C (nested ANOVA: $df = 4$, $F = 2.6161$, $p = 0.045$; post-hoc Dunnett's test no difference to control). The heart rate of the fish exposed at 11 °C was about 20 beats per minute higher on the average than in fish exposed at 7 °C. The weight and length of fish exposed to 1,000 μg/L citalopram was significantly lower compared to the control at both temperatures (Table 1) (11 °C weight: nested ANOVA: $df = 4$, $F = 12.8137$, $p < 0.001$; post-hoc Dunnett's test [0 μg/L|1000 μg/L] $p < 0.001$; 11 °C length: nested ANOVA: $df = 4$, $F = 13.1786$, $p < 0.001$; post-hoc Dunnett's test [0 μg/L|1000 μg/L] $p < 0.001$; 7 °C weight: nested ANOVA: $df = 4$, $F = 9.7415$, $p < 0.001$; post-hoc Dunnett's test [0 μg/L|1000 μg/L] $p < 0.001$; 7 °C length: nested ANOVA: $df = 4$, $F = 22.0216$, $p < 0.001$; post-hoc Dunnett's test [0 μg/L|1000 μg/L] $p < 0.001$).

### Swimming behaviour during exposure

We could show that citalopram had an effect on the stay of the fish in the upper half of the aquaria at both temperatures. Significantly more fish exposed to 1,000 μg/L citalopram

stayed close to the water surface compared to the control fish at both temperatures. Fish exposed to 1 and 10 µg/L citalopram concentrations at 7 °C stayed slightly more often in the upper half of the aquaria than control fish (Table 1) (11 °C: Generalized Linear Mixed Model, $df = 4$, $F = 12.4141$, post-hoc Dunnett's [0 µg/L|1000 µg/L] $p < 0.001$; 7 °C: Generalized Linear Mixed Model, $df = 4$, $F = 106.9664$, post-hoc Dunnett's [0 µg/L|1µg/L] $p = 0,027$, [0 µg/L|10 µg/L] $p = 0.017$, [0 µg/L|1000 µg/L] $p < 0.001$).

*ASMD*
Overall, fish exposed at 11 °C swam further and faster in the ASMD than fish exposed at 7 °C. Furthermore, the citalopram treatments also revealed an effect on the total distance moved and the mean velocity during the recordings in the ASMD. Fish exposed to 1,000 µg/L citalopram at 11 °C swam significantly less and slower than control fish (distance moved: nested ANOVA: $df = 4$, $F = 4.7551$, $p = 0.002$; post-hoc Dunnett's test [0 µg/L|1000 µg/L] $p = 0.001$). Also, fish exposed at 7 °C showed significantly less total distance moved and had a lower mean velocity when exposed to 100 µg/L or 1,000 µg/L citalopram, compared to control fish (Table 1, Figs. 1A, 1D) (distance moved: nested ANOVA: $df = 4$, $F = 7.8214$, $p < 0.001$; post-hoc Dunnett's test [0 µg/L|100 µg/L] $p = 0.021$ [0 µg/L|1000 µg/L] $p < 0.001$). Furthermore, it can be seen that control brown trout larvae have an increased time of no movement at the beginning of the experiment, which decreases overtime. Contrary mean velocity at the beginning of the experiment is lower and increases with time and has its peak at 12 min. In contrast brown trout larvae exposed to the highest treatment have a constant lower mean velocity over time and a constant longer time of no movement over recordingtime (Figs. 1B–1C, 1E–1F). Brown trout larvae exposed to 100 and 1,000 µg/L at 7 °C show significant decreased mean velocity and increased time of no movement over time (mean velocity: Linear mixed model: $df = 4;10$, $F = 5.6897$, $p = 0.012$, post-hoc Dunnett's [0 µg/L|100 µg/L] $p = 0.024$, [0 µg/L|1000 µg/L] $p < 0.001$; no movement: Linear mixed model: $df = 4;10$, $F = 5.4797$, $p = 0.013$, post-hoc Dunnett's [0 µg/L|100 µg/L] $p = 0.041$, [0 µg/L|1000 µg/L] $p < 0.001$). Brown trout larvae exposed at 11 °C do not show significant difference, though a statistical trend is visible (mean velocity: Linear mixed model: $df = 4;10$, $F = 2.7222$, $p = 0.091$, no movement: Linear mixed model: $df = 4;10$, $F = 3.1072$, $p = 0.066$).

## Experiment with juvenile brown trout
No mortality occurred during the experiment. Weight and length were significantly lower in fish exposed to 1,000 µg/L citalopram compared to control fish (weight: nested ANOVA: $df = 4$, $F = 3.2964$, $p = 0.001$; post-hoc Dunnett's test [0 µg/L|1000 µg/L] $p = 0.023$; length: nested ANOVA: $df = 4$, $F = 4.6661$, $p = 0.002$; post-hoc Dunnett's test [0 µg/L|1000 µg/L] $p = 0.019$).

*Swimming during exposure*
There was a strong effect of the highest citalopram concentration on the swimming behaviour of fish: About 25% of the 1,000 µg/L citalopram-treated fish stayed in the upper half of the aquaria, in contrast to the control and other treatments, where no fish stayed
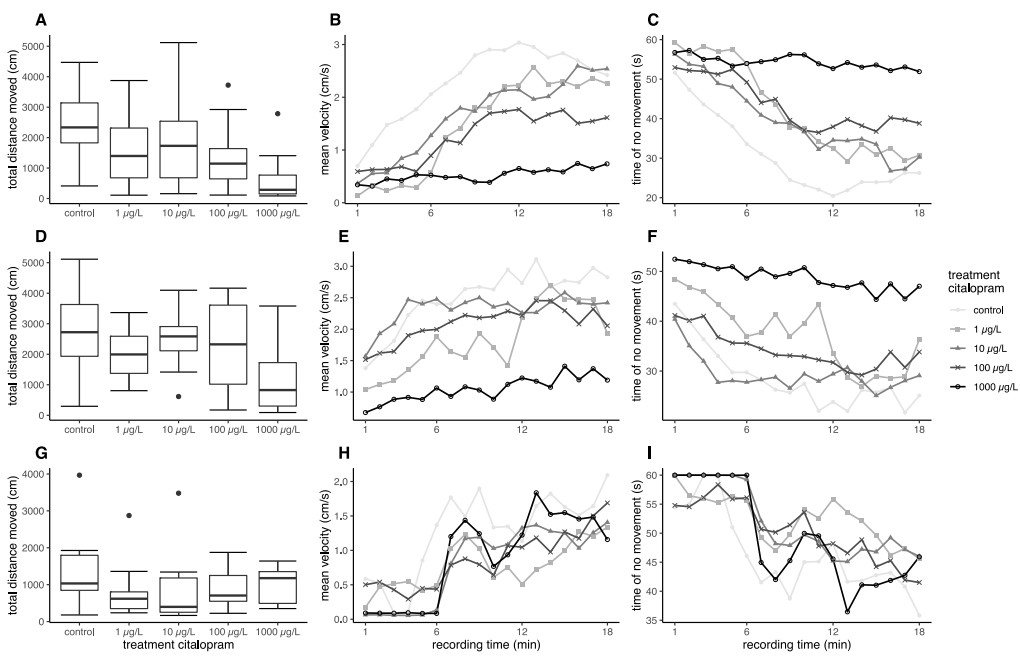

**Figure 1** **Results for brown trout in the ASMD.** Results of total distance moved are shown as boxplot (A, D, G), Results of mean velocity (B, E, H) and time of no movement (C, F, I) are represented as dots, with depicted mean values over time. (A–C) Results for brown trout larvae exposed at 7 °C, (D–F) Results for brown trout larvae exposed at 11 °C, (G–H) Results for juvenile brown trout.

in the upper aquaria part (Generalized linear mixed model, $df = 4$, $F = 7.3259$, post-hoc Dunnett's [0 µg/L|1000 µg/L] $p = 0,0011$) (Table 2).

### ASMD

In the artificial swimming measurement device, neither the total distance moved nor the mean velocity of the exposed fish differed significantly to the control (Table 2, Fig. 1G) (distance moved: nested ANOVA: $df = 4$. $F = 1.0846$. $p = 0.382$; velocity: nested ANOVA: $df = 4$, $F = 1.0846$, $p = 0.382$). Mean total distance moved and averaged mean velocity of the exposed fish were about 70% of the fish from the control; however, this was not significant. The behaviour pattern over time does not differ strongly between control and treated fish. At the beginning juvenile brown trout have a reduces mean velocity, which increases over time. Opposed to this time of no movement is decreasing over time (Figs. 1H–1I). Statsitical difference between exposed and control fish could not be revealed with regard to mean velocity and no movement over time (mean velocity: Linear mixed model: $df = 4;10$, $F = 0.4324$, $p = 0.782$; no movement: Linear mixed model: $df = 4;10$, $F = 0.4902$, $p = 0.743$).

### Cortisol

Tissue cortisol concentrations did not differ between exposed and control fish. The citalopram exposed fish did not show significant differences between the treatments (Table 2, Fig. 2) (Linear Mixed Model: $df = 4,68.638$, $F = 3.7625$, $p = 0.008$, post-hoc

**Table 2 Results for juvenile brown trout exposed to citalopram.**

| Treatment (μg/L) | 0 | 1 | 10 | 100 | 1000 |
|---|---|---|---|---|---|
| Mortality (%) | $0 \pm 0$ | $0 \pm 0$ | $0 \pm 0$ | $0 \pm 0$ | $0 \pm 0$ |
| Weight (g) | $2.75 \pm 0.84$ | $2.85 \pm 0.85$ | $2.86 \pm 1.16$ | $2.74 \pm 0.92$ | **$2.17 \pm 0.53$ \*** |
| Length (cm) | $6.41 \pm 0.65$ | $6.59 \pm 0.61$ | $6.46 \pm 0.71$ | $6.50 \pm 0.75$ | **$5.93 \pm 0.49$ \*** |
| Aquaria photographed | 2 | 2 | 1 | 2 | 2 |
| Fish in upper aquaria half (%) | $0 \pm 0$ | $0 \pm 0$ | $0 \pm 0$ | $0 \pm 0$ | **$25.42 \pm 19.11$ \*\*** |
| ASMD: total distance moved (cm) | $1371 \pm 1057$ | $847 \pm 785$ | $878 \pm 1004$ | $939 \pm 571$ | $980 \pm 473$ |
| Cortisol content in fish extract (ng/mL) | $19.06 \pm 14.80$ | $15.51 \pm 7.48$ | $12.50 \pm 9.58$ | $20.76 \pm 15.14$ | $23.66 \pm 17.81$ |
| Aqueous citalopram concentration (μg/L) | <LoD | $1.41 \pm 0.22$ | $9.20 \pm 0.59$ | $81.51 \pm 2.39$ | $864.93 \pm 51.54$ |
| Tissue citalopram concentration (μg/g) (wet weight) | <LoD | $8.2 \pm 4.37$ | $38.3 \pm 30.71$ | $340.63 \pm 124.74$ | $2966.83 \pm 1556.77$ |

Notes.

Data are shown as arithmetical mean ± standard deviation.

Asterisks indicate significant differences to the respective controls (\*$p < 0.05$; \*\*$p < 0.01$; \*\*\*$p < 0.001$).

LoD, limit of detection.

Dunnett's Test revealed no difference between control and treatments). However, there was a significant increase in tissue cortisol content in fish tested in the ASMD (mean cortisol content: $26.66 \pm 18.57$) compared to fish not tested in the ASMD (mean cortisol content: $15.22 \pm 10.16$) (Linear Mixed Model: $df = 1,95.378$, $F = 16.7132$, $p < 0.001$) (Fig. 2).

# DISCUSSION

This study shows that citalopram affects the swimming behaviour and growth of brown trout in different life stages. Effect concentrations were close to citalopram concentrations measured in wastewater effluents (*Fick et al., 2009*; *Larsson, De Pedro & Paxeus, 2007*; *Nodler et al., 2010*; *Vasskog et al., 2006*).

## Bioconcentration

It has been shown that citalopram can accumulate in the liver, kidney and brain of fish (*Grabicova et al., 2017*; *Grabicova et al., 2014*). In the present study, we analysed tail fin tissue samples (muscle) since all other organs were used for biomarker analyses. The obtained data made evident that even early life stages of brown trout but, more intensely, juveniles accumulate citalopram in their muscle tissue. In both life stages, citalopram tissue concentrations rose slightly with increasing exposure concentrations of 1 to 100 μg/L (Tables 1 and 2). When comparing the aqueous with the tissue concentration there is a linear relationship between exposure concentrations and internal concentration (Supplemental material). Citalopram concentrations in fish exposed at 7 °C (135 days) were about 3 times higher than those in fish exposed at 11 °C (107 days), possibly due to the longer exposure time of about 4 weeks. The muscle tissue concentration of juvenile brown trout was about 10 times higher than the muscle tissue concentration of brown trout larvae possibly based on a more intense citalopram uptake due to the ongoing development of the gastro-intestinal system and/or gills of larvae. *Sackerman et al. (2010)* showed a bioconcentration of $115 \pm 37$ ng/g citalopram in the brain and $193 \pm 33$ ng/g in the muscle tissue of zebrafish exposed to 24.3 μg/L for only 3 min. Brown trout

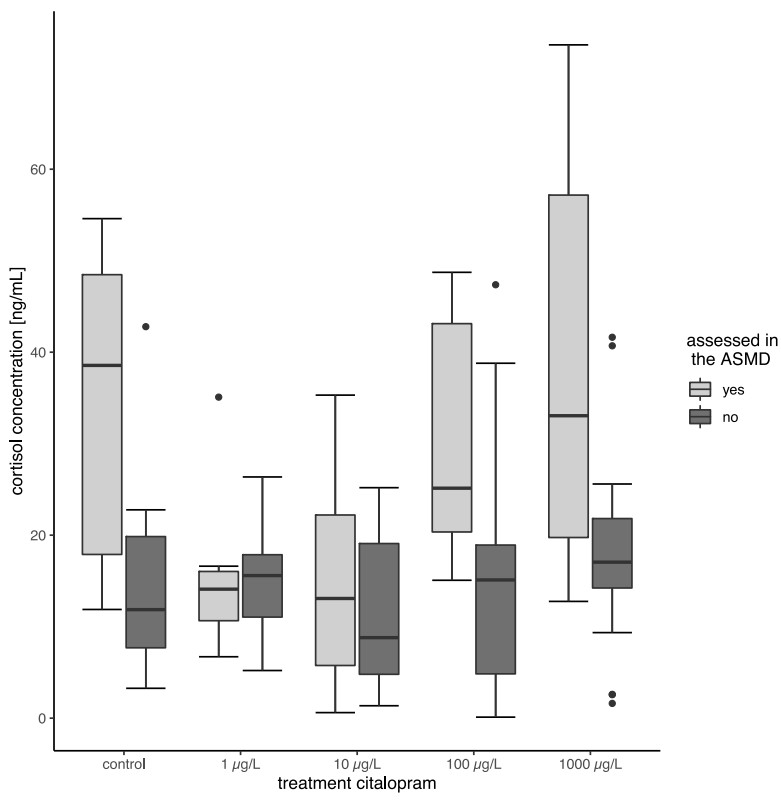

**Figure 2 Cortisol content of juvenile brown trout.** Cortisol content of the fish assessed in the ASMD compared to whose swimming behaviour was not assessed in the ASMD. Results are shown as boxplots.

exposed to an effluent-dominated stream (citalopram water concentrations: 4.5–58 ng/L) in the Czech Republic showed citalopram concentrations up to $31 \pm 11$ ng/g in liver and $164 \pm 19$ ng/g in kidney tissue; however, no citalopram could be detected in brain and muscle tissue (*Grabicova et al., 2017*). Likewise, rainbow trout exposed to the effluent of a Swedish wastewater treatment plant (citalopram water concentration: $260 \pm 60$ ng/L) had most citalopram in the liver and brain, with concentrations of $12 \pm 5$ ng/g and $2.2 \pm 1.3$ ng/g, respectively. In contrast, no citalopram was detected in plasma and muscle tissue (*Grabicova et al., 2014*). The reason for finding citalopram being accumulated in muscle tissue in our study might either be due to the fact that the exposure time was longer and the test concentrations were higher than those used in other studies (*Grabicova et al., 2017*; *Grabicova et al., 2014*; *Sackerman et al., 2010*) or result from a diverging accumulation pattern in brown trout compared to other fish species (*Du et al., 2016*).

## Mortality

Mortality of larvae and juvenile fish was not influenced by citalopram exposure. Although the overall mortality increased to 29% in brown trout larvae exposed at 11 °C. This higher mortality relates to the fact that brown trout larvae exposed to 11 °C had a higher metabolism and that the beginning of exogenous feeding in salmonids is associated with higher mortality risks (*Klemetsen et al., 2003*). Also in a study by *Kellner et al. (2016)*, in

which sticklebacks have been exposed for 21 days to 1.5 and 15 μg/L citalopram, no differences in mortality were found. Likewise, time to hatch was not affected by citalopram. This is in accordance with other studies that did not reveal differences in hatching success and time for zebrafish or Japanese medaka exposed to the SSRI fluoxetine (*Foran et al., 2004*; *Wu et al., 2017*). With regard to the developmental parameter heart rate of brown trout larvae, no effect was seen in fish exposed to either temperature and any citalopram concentration corroborating finding for other SSRIs by *Airhart et al. (2007)*, who exposed zebrafish larvae to 1,39 mg/L of the SSRI fluoxetine.

## Fish growth

In our experiment, we visually observed more food leftovers in the exposure tanks with 1,000 μg/L citalopram, which allowed us to conclude that there was a lower food intake in fish exposed to 1,000 μg/L citalopram; however, a quantification of this effect was not possible. Known side effects of citalopram are anorexia and weight loss in humans (Information of *HEXAL (2012)*) and published data on fish revealed a decreased food intake in sticklebacks exposed to 0.15 μg/L citalopram (*Kellner et al., 2015*). Reduced food intake in fish was also shown for two other antidepressants, sertraline and fluoxetine, in European perch and goldfish (*Hedgespeth, Nilsson & Berglund, 2014*; *Mennigen et al., 2010*). In contrast three-spined stickleblacks exposed for 30 days to 1.5 μg/L citalopram with a subsequent 185 days recovery phase showed increased attacks on a chironomid larvae (*Kellner et al., 2017*). Mechanistically, the increased swimming activity and the conceivably decreased food intake of the fish make the observed decrease in weight and length of the fish exposed to 1,000 μg/L citalopram is reasonable to result from exposure to citalopram, especially as exposed fish were in a period of intense growth. Decreased weight has also been shown in goldfish and decreased length in zebrafish exposed to 54 and 10 μg/L of the SSRI fluoxetine, respectively (*Mennigen et al., 2010*; *Wu et al., 2017*).

## Behaviour during exposure

Control brown trout preferably stayed at the bottom of the tank when kept in the exposure tanks (Tables 1 and 2). Fish exposed to 1,000 μg/L citalopram showed an increased preference for the upper half of the aquaria independent of their life stage. In general, the test design used in the present study for this parameter implies parts of the novel tank diving test, the scototaxis test and shoaling. *Stewart et al. (2012)* described the novel tank diving test for the measurement of anxiety, where single zebrafish are placed in tanks. Time spent in the upper aquaria portion is recorded along with other parameters like the number of transitions into the upper aquaria portion or number of freezing bouts. When the fish in a novel tank stay in the lower aquaria part (geotaxis) it is a sign for anxiety behaviour and in contrast the transition into the upper aquaria part is a sign of boldness and therefore anti-anxiety behaviour (*Stewart et al., 2012*). *Maximino et al. (2010)* described the scototaxis test, where fish are transposed into a novel tank and can freely swim into the dark or bright zone. Preferring the dark part of the aquaria and avoiding the brighter side is a clear sign of anxiety (*Maximino et al., 2010*). Furthermore, *Stewart et al. (2012)* described fish with lower anxiety to have a greater tendency to break away from

the shoal in zebrafish. It is obvious that in our case swimming behaviour of the shoal of brown trout was assessed and not individual swimming behaviour. Therefore, the stressor not being in a shoal, like in the novel tank and scototaxis test is not present in our case. Additionally, brown trout were not transposed into novel tanks, but the tanks were covered the most of the exposure time with black foil, only when the photos were taken, the foil was removed, which lead to a higher illumination from the top. Overall an increased stay in the upper aquaria part is a preference for the brighter illuminated water column and water surface in contrast to the darker aquaria bottom. Additionally, these fish recede from the shoal at the bottom of the tank. In our experiments, up to 80% brown trout larvae and 25% of juveniles exposed to 1,000 µg/L citalopram stayed in the upper aquaria section compared to the control animal. This can be explained by decreased anxiety and an altered swimming behaviour characterised by a higher activity of the exposed fish (*Maximino et al., 2010*; *Stewart et al., 2012*). The stronger effect of citalopram on the vertical distribution of the brown trout larvae can be explained by a 5 times longer exposure time compared to the juvenile individuals. Also, different sensitivities of the life stages can come into play, and has already be shown for diclofenac (*Schwarz et al., 2017*). The significant difference in time spent in the upper aquaria part of the brown trout larvae exposed to 1 µg/L citalopram at 7 °C could also be due to the fact that only one of the three replicate aquaria was photographed and therefore a single individual has a higher impact on the relative number spent in the upper aquaria part. For this reason and the inherent variation in this setup for behaviour measurement the biological relevance of the slight effect in the 1 and 10 µg/L treatment has to be confirmed with a bigger sample size. An anxiolytic effect of citalopram in the novel tank test was also shown for other fish species like Endler's guppies, three-spined sticklebacks and zebrafish, even at decidedly lower concentrations of citalopram down to 1.5 µg/L, which all spent more time in the upper aquaria part during the novel tank test (*Kellner et al., 2016*; *Olsen et al., 2014*; *Sackerman et al., 2010*). Furthermore, *Kellner et al. (2017)* discovered increased transitions to the bright side in sticklebacks exposed to 1.5 µg/L citalopram for 30 days with subsequent 120 days recovery phase. But latency to first transition and time spent in the brighter zone was not influenced. Also sticklebacks exposed did not show any difference to control fish in the novel tank test, except for decreased acitivty (*Kellner et al., 2017*). *Kellner et al. (2016)* also observed increased swimming activity in fish exposed to 1.5 µg/L citalopram. This effect has not only been shown in response to citalopram but has also been found for other antidepressants like fluoxetine or amitriptyline, which seem to reduce anxiety and increase the stay of the fish in the upper part of the aquarium (*Demin et al., 2017*; *Henry & Black, 2008*; *Meshalkina et al., 2018*)..

## Behaviour in a stressful environment and cortisol measurements

In contrast to the increased swimming activity under minor stress conditions, brown trout larvae exposed to citalopram showed a decreased swimming activity in the stressful artificial swimming measurement device (ASMD). This effect can be due to the anxiolytic and soothing effect of the antidepressant. Based on the measurements of the tissue cortisol level, it is evident that the ASMD creates a rather stressful environment for the fish (Fig.

2) (*Wendelaar Bonga, 1997*). The transfer of fish into an ASMD leads to a stress reaction in larvae. This stress reaction is characterised by freezing behaviour at the beginning followed up by increased velocity and escape behaviour (Fig. 1). Having a look on the time dependant velocity and time of no movement, it is clear that the control larvae are swimming faster and have reduced time of no movements over recording. In contrast larvae exposed to 1 mg/L citalopram do not show this pattern, but have a constant low mean velocity and longer time of no movement over the total recording time. A similar pattern can be seen in fish exposed to 100 µg/L citalopram but not as distinct as in fish exposed to 1 mg/L. Since citalopram is an anxiolytic drug and reduces anxiety, fish are reasonably soothed when exposed to 100 µg/L or 1,000 µg/L citalopram as they swam slower than control fish over the total recording time. Therefore, the total distance they moved and their mean swimming velocity were lower than that of the controls. Additionally, the time dependant behaviour pattern is not given in fish exposed to 1 mg/L citalopram. It is also possible the reduced mean velocity and increased time of no movement is due to a sedative effect of the antidepressant. Furthermore, it cannot be excluded, that the increased time of no movement of fish exposed to 1 mg/L citalopram represents freezing and therefore anxiogenic behaviour. Nevertheless, with regard to the results of the behaviour during the exposure, it is more likely that the effect is due to a reduction of anxiety in brown trout larvae which resulted in calmed behaviour. This reduction in activity due to citalopram in the ASMD could not be seen in juvenile fish. Neither total distance moved nor the pattern over time differed between the treatment and the control animals. Nevertheless, the total distance moved and the mean velocity of swimming in all citalopram-treated juveniles was 70% lower than in controls, but these differences were not significant. Similar to the effect on citalopram on vertical distribution patterns during exposure, behavioral changes in the ASMD were more pronounced in exposed larvae than in juveniles. As previously mentioned this might be the result of the longer exposure of larvae or different sensitivities between life stages (*Schwarz et al., 2017*). It has been reported previously that antidepressants reduce swimming activity. Three-spined sticklebacks exposed to 1.5 µg/L citalopram for 30 days at a developmental stage with subsequent 160 days recovery phase showed a decrease in swimming activity (*Kellner et al., 2017*). Zebrafish embryos exposed to either 1 mg/L venlafaxine or 3 mg/L sertraline showed reduced swimming behaviour in the zebrabox (*Sehonova et al., 2019*). With regard to the sedative effect of antidepressants, for example, western mosquitofish exposed to 0.5 µg/L fluoxetine over 91 days showed increased lethargy similar to fish exposed to 53 µg/L for 7 days (*Henry & Black, 2008*). Furthermore, guppies exposed to 16 ng/L fluoxetine for 28 days showed increased freezing time and time spent under cover (*Saaristo et al., 2017*). Male bluehead wrasse injected intraperitoneal with 6 µg/g bw fluoxetine showed decreased activity levels (*Perreault, Semsar & Godwin, 2003*). But also meagre exposed to 20 µg/L venlafaxine showed reduced swimming activity (*Maulvault et al., 2018*). But it could also be shown that antidepressants reduces anxiety in exposed fish: *Painter et al. (2009)* showed a decreased escape behaviour in fathead minnow larvae exposed to 250 ng/L of the SSRI fluoxetine, resulting in reduced swimming velocity. Likewise, fathead minnow larvae exposed to the serotonin and noradrenalin reuptake

inhibitor (SNRI) venlafaxine revealed reduced anxiety, indicated by a reduced escape response (*Painter et al., 2009*).

## Calculated plasma concentrations

Therapeutic human plasma concentrations in patients treated with doses of 20–60 mg citalopram per day are $117 \pm 95$ µg/L (*Le Bloc'h et al., 2003*). In contrast, *Schreiber et al. (2011)* reported a maximum blood plasma concentration of only 21.1 µg/L in patients after drug administration with a maximum daily dose of 60 mg. Considering measured human therapeutic plasma concentrations of citalopram, calculated plasma concentrations in fish exposed to 1, 10, 100 and 1,000 µg/L at pH 8 with the fish plasma model (Table S8) (*Fu, Franco & Trapp, 2009*; *Huggett et al., 2003*; *Schreiber et al., 2011*) revealed that the calculated concentrations in fish exposed to 100 µg/L citalopram or higher exceeded the human therapeutic plasma concentrations according to *Le Bloc'h et al. (2003)*. When referred to the human plasma concentrations in the study conducted by *Schreiber et al. (2011)*, when even exposed to 10 µg/L fish plasma concentrations would exceed the concentrations in human plasma. However, *Holmberg et al. (2011)* showed that 2 out of 5 rainbow trout exposed to 10 µg/L citalopram revealed a plasma citalopram concentration of 0.044 µg/L and 0.08 µg/L after exposure for only 24 h. The lack of behavioural effects in the study of *Holmberg et al. (2011)* and also our results obtained for the lower treatments 1 µg/L and 10 µg/L citalopram, suggests that citalopram plasma concentrations in fish below human therapeutic plasma concentrations do not to affect the fish. Though, our results on behaviour and growth of brown trout provide evidence that citalopram plasma concentrations in fish higher than human therapeutic plasma concentrations can have severe impact on brown trout in different life stages.

## CONCLUSION

Our results clearly show that citalopram affects brown trout according to its mode of action known for humans. Under stressful conditions, fish showed reduced swimming behaviour when exposed to at least 100 µg/L citalopram. Furthermore, in the exposure tanks, an increased swimming activity during exposure was observed for fish exposed to 1 mg/L citalopram, which can be linked to the anti-depressant effect of the drug. The behavioural changes were stronger in early life stages, which could be associated with the longer exposure time in larvae compared to juveniles, but also differences in sensitivity between life stages may play a role. In addition, side-effects of the antidepressant known from human applications could be detected, like reduced weight and length, in both juvenile brown trout and brown trout larvae exposed to 1,000 µg/L citalopram. Our results confirm similar findings for citalopram exposure to those reported for other aquatic species. To conclude, citalopram, as a widely distributed drug, severely alters the behaviour and growth of brown trout in different life stages, at concentrations higher than current environmentally relevant levels. And the 10 times stronger accumulation of citalopram in juveniles makes evident that an increase in surface water concentration of citalopram could have severe impact on specific life stages of fish. Nevertheless, considering safety factors up to $10^3$ that have to be included in environmental risk assessment and also

additive effects of pharmaceuticals affecting similar pathways, citalopram is far from being an environmentally safe pharmaceutical and has to be considered carefully with respect to risk for the aquatic environment.

## ACKNOWLEDGEMENTS

Particular thanks go to Thomas Braunbeck, Heidelberg University, for the coordination of this project. Furthermore, the authors thank Stefanie Jacob, Stefanie Krais, Elisabeth May, Katharina Peschke, Lukas Reinelt, Hannah Schmieg and Sabrina Wilhelm for help in the laboratory and technical assistance, and Stefanie Dietz for comments on the manuscript. Futhermore, thanks go to Nils Anthes and Simon Schwarz for statistical advice. Language check was conducted by Proof-Reading-Service.com.

### Funding

This study is part of the project Effect-Net (Effect Network in Water Research), which is part of the Water Research Network Baden-Württemberg (Wassernetzwerk Baden-Württemberg) and funded by the Ministry for Science, Research and Arts of Baden-Württemberg. Carolin Huhn received support from the Excellence Initiative, a jointly funded program of the German Federal and State governments, organized by the German Research Foundation (DFG). The authors also received support from the Open Access Publishing Fund of University of Tübingen. The funders had no role in study design, data collection and analysis, decision to publish, or preparation of the manuscript.

### Grant Disclosures

The following grant information was disclosed by the authors:
Effect-Net (Effect Network in Water Research).
Water Research Network Baden-Württemberg (Wassernetzwerk Baden-Württemberg).
Ministry for Science, Research and Arts of Baden-Württemberg.
German Federal and State governments, organized by the German Research Foundation (DFG).
Open Access Publishing Fund (University of Tübingen).

### Competing Interests

The authors declare there are no competing interests.

### Author Contributions

- Michael Ziegler conceived and designed the experiments, performed the experiments, analyzed the data, prepared figures and/or tables, authored or reviewed drafts of the paper, and approved the final draft.
- Sarah Knoll and Selina Tisler analyzed the data, authored or reviewed drafts of the paper, and approved the final draft.

- Heinz-R. Köhler, Carolin Huhn, Christian Zwiener and Rita Triebskorn conceived and designed the experiments, authored or reviewed drafts of the paper, contributed reagents/materials/analysis tools, and approved the final draft.

### Animal Ethics
The following information was supplied relating to ethical approvals (i.e., approving body and any reference numbers):

Animal welfare committee of the Regional Council of Tübingen, Germany approved the study (authorisation ZO 2/16).

### Data Availability
Data is available at Effect-Net under "brown trout larvae exposed to citalopram" and "juvenile brown trout exposed to citalopram": https://effectnet-seek.bioquant.uni-heidelberg.de/investigations/9.

### Supplemental Information
Supplemental information for this article can be found online at http://dx.doi.org/10.7717/peerj.8765#supplemental-information.

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
