# Peer review of "Impact of the antidepressant citalopram on the behaviour of two different life stages of brown trout"

_PeerJ, doi:10.7717/peerj.8765_

## Round 0.1 · original submission · Major Revisions

There are 2 really thorough reviews by patient individuals interested in your work. Please respond to the reviews and modify your manuscript accordingly.

Reviewer 1 ·

Basic reporting

In general, the results and statistics are well presented. However, the writing is at times difficult to follow and the wording ambiguous. I have made a few comments to help. In places, arguments are not justified or supported by references, please ensure appropriate supporting references are added. Line by line feedback is copied below.

Abstract background: In the background section, switching between the frequency of human prescriptions vs environmental presence of antidepressants is potentially confusing. I would recommend a more linear framework. There are also a few small typos. I would recommend the following changes… “Over the last two decades, there has been a constant increase in prescription rates of antidepressants. In parallel, neuroactive pharmaceuticals are making their way into aquatic environments at increasing concentrations. Among the antidepressants detected in the environment citalopram, a selective serotonin reuptake inhibitor, is one of the most prevalent. Given citalopram is specifically designed to alter mood and behaviour in humans, there is growing concern it can adversely affect the behaviour on non-target wildlife.”

Abstract methods: For the method section, there is no mention of what endpoints were actually measured at each life stage, or the nominal concentrations used. Both should be added. Further, I would recommend removing “Real water concentrations well reflected nominal concentrations.”, this is not necessary in the methods section. For example…“In our study, brown trout were exposed to environmentally relevant citalopram concentrations (nominal concentrations 1, 10, 100, 1000 ug/L) at two different life stages, a larvae exposure (eyed ova stage until 8 weeks post yolk sac consumption) and a juvenile exposure (X age for a 4-week exposure). At both stages, we measured…XX.”
Results: In the results section, you may consider replacing “sojourn” it is not a very commonly used term (as far as I have seen). Instead… “both larvae and juvenile fish exposed to the highest test concentration of citalopram (1 mg/L) had higher swimming activity and decreased anxiety (i.e. spent longer in the upper part of the aquaria) compared to control fish.

Line 53: “in rising product types” do you mean an increasing number of different psychoactive drugs? If so, perhaps it could be re-worded to make this clearer.

Line 56–57: “Within the group of antidepressant drugs, the selective serotonin reuptake inhibitors (SSRI) play a major role (Schwabe and Paffrath, 2016).” — This is a little ambiguous, play a major role in what? Treating patients? Please clarify.

Line 63: Add a comma after Germany.

Lines 67–69: “In wastewater treatment plant effluents, citalopram concentrations range from 44 ng/L to 840 μg/L in the effluent of drug manufacturers (Himmelsbach et al., 2006; Larsson et al., 2007; Silva et al., 2014).” — This sentence is ambiguous; do you is this only in effluent of drug manufacturers or more broadly in wastewater treatment plant effluents. Please clarify.

Lines 74–82: It needs to be more clear which results refer to which studies when comparing and contrasting past studies. It should also be made more clear which results are in agreement and which are not. For example, “At present, the impacts of citalopram on anxiety-related behaviour of fish have been mixed (REF all studies). X studies reported a decrease in anxiety behaviour at X concentrations with X length exposure… whereas, X studies reported…” At the moment it does appear that there are many discrepancies.

Line 79 and 81: Please give exact concentration instead of low ug/L.

Line 88–89: “Due to these diverse outcomes, it is important to assess different endpoints in different life stages to detect the diverse effects that citalopram can have on aquatic organisms” — With the above examples it is really not clear which are divergent and why? Across these studies there are many different concentrations, dose duration, and species could you make this clearer as you make these comparisons. I fear that you may be exaggerating inconstancies across studies, which makes the impacts of citalopram on fish behaviour seem weak and unrepeatable. Please make these comparisons more clearly, as to which results are contrary to each other, as well as including concentrations, dose duration, and species for each.

Line 100: Again, I think sojourn is not used very frequently, it might help to replace it with simpler language.

Line 123: Sub-heading is not bolded like the others.

Line 125–126: “both 7°C and 11°C, in order to reveal influences of temperature on the effects investigated” – This is an interesting point of difference of your study, it would be good to highlight this objective in the introduction and abstract.

Line 225: “Artificial swimming measurement device” – This is a form of novel tank test, right? I would change the name to novel tank test throughout as most readers will be familiar with this terminology.

Experimental design

I believe there were some issues with experimental design in the project. Most importantly, the methods used to measure 'Swimming behaviour during exposure', as well as low sample sizes more generally. I have detailed my thoughts in line by line feedback below.

Line 136–138: “At 7 days post-hatch, the heart rate of 5 individuals of each control and the highest concentration tank was measured and, whenever a difference was revealed, the other treatments were also assessed.” – I would not recommend such a protocol in future experiments; SSRIs are increasingly reported to have non-monotonic dose-response relationships. Therefore, you may miss an effect at your low dosage that was not reflected at the higher dosage. As one example, in the highest dosage you may see phenotypes of a toxic effect of the chemical, which has opposite effects to what you would see based on the therapeutic action of the chemical. Obviously, it is not possible to change this protocol now, but I would recommend that it is avoided in future experiments.

Line 141–143: “Here, 5 fish from each tank were transferred into small glass aquaria and swimming behaviour was recorded for 18 minutes. Terminally, fish were sampled 8 weeks after yolk sac consumption.” —A sample of 5 fish is not an adequate sample size to draw a comprehensive behavioural conclusion on the effect of low dose chemical exposure. All conclusion from this behavioural data should be very circumspect and should acknowledge the limited sample size and thus the power of the results.

Line 218–220: “For experiments with brown trout larvae, 3 pictures were taken per day of each photographable tank at an interval of 5 minutes, from one-week prior to the experiment until sampling (Apr 03 – Apr 13, 2017).” — Please justify that this method is robust enough to capture changes in swimming behaviour. This is a novel method, therefore, extra justification of whether it can capture changes in swimming behaviour is necessary. My ‘knee jerk’ reaction is that it is not sufficient to detect changes in swimming behaviour, as it captures a total of only 27 frames of the swimming behaviour over a week, in video terms, this is less than a second (i.e. 30 fps) of behaviour over a week. Fish move rapidly and will change their behaviour based on an observer being present (as they would have been during the photo). I think more detailed information on fish swimming behaviour is required than that capture in this method. This is further compounded by small sample size.

Line 228: “and five brown trout larvae were placed in there” – Again, 5 fish is not an adequate sample size to draw behavioural conclusions. Indeed, looking at table one I note a huge amount of variability in total distance moved in all fish groups. Subtle effect at the lower dosages could have been missed due to low statistical power.

Validity of the findings

The arguments for the behavioural effects reported need to be supported with references. In addition, the small sample size of this study make it difficult to make concrete conclusions on the effects of low-dose exposure.

Line 449–450: “higher locomotor activity of the exposed fish (Stewart et al., 2012)” — I do not believe can you conclude this if you didn’t measure swimming velocity (using the photographic method). Spending time in the upper part of the tank doesn’t necessary mean they had higher swimming velocities.

Line 456–458: “For this reason and the inherent variation in this setup for behaviour measurement the biological relevance of the slight effect in the 1 and 10 μg/L treatment has to be confirmed with a bigger sample size.” — This is true for all the behavioural results in this study, whether they are ‘positive’ or ‘negative’ results.

Line 475–480: “The transfer of fish into an ASMD leads to a stress reaction in fish, which basically results in a startle reflex and increased escape behaviour and thus, an increase in locomotor activity. Since citalopram is an anxiolytic drug and reduces anxiety, fish are reasonably soothed when exposed to 100 μg/L or 1000 μg/L citalopram as they swam slower and less bustling than control fish. Therefore, the total distance they moved and their mean swimming velocity were lower than that of the controls.” – I understand the logic of this argument, however, it could just as easily be argued that decreased anxiety would increase activity in a novel stressful environment. A fish that remains more activity and exploratory in a novel environment stressful environment is typically considered bolder and less anxiety in the behavioural literature (as you have argued in the section above). If these fish are preforming a stress escape/startle response, it should not last the whole trial, instead, you would expect that they return to normal activity levels over the duration of the experiment. Perhaps looking at your velocity data over time will give some insights. Alternatively, is there a study which has confirmed your argument in which you can cite.

Additional comments

The work you are doing is very important. We need to generate more data on the impacts of low-dose pharmaceutical exposure on sublethal endpoints. However, I would encourage the authors to use larger sample sizes in future experiments, the effects of these chemicals can be 'messy' (and understandably so given their mode of action), large statistical power is essential to identify the realised effects of these drugs. If the authors make a point of explaining these issues forthrightly and acknowledge the limitation of the behavioural conclusions—specifically at the lower dosages that had high within-group variability—the data generated would be useful to the wider ecological community.

Reviewer 2 ·

Basic reporting

The article is overall well written and easy to understand, although the authors introduce new terms that are not commonly used in the field (e.g. sojourn, ASMD). I have added suggestions to replace these with more commonly used terminology.

Literature references are appropriate and a good overview of the studies on citalopram in fish is provided in the introduction. However, in the discussion only the studies confirming the results from the present paper are reitterated, while the discussion section would benefit from a more balanced review of how the current results relate to previous studies.

The article is well structured. However, I request more information in the form of tables and figures on the behavioral parameters from the "ASMD". The current conclusions are not supported by only presenting average values of distance moved and velocity. Also ammonium levels are missing from the water conditions table (supplement). Raw data is currently not shared.

Experimental design

The article fills a knowledge gap as studies effects of citalopram pollution on fish have reported diverging results, and citalopram is one of the most commonly used SSRIs in Germany.

The experiment is generally well performed, although there is room for improvement with regards to the reporting of the egg exposure experimental design, reporting of behavioral measurements and reporting of ammonium levels (considering the high fish density and infrequent water changes).

The experimental design (exposure setup) and statistical analyses are sound.

Validity of the findings

Impact and novelty are assessed and meaningful replication is encouraged.

Not all underlying data has been provided (see below).

My main 2 objections to the conclusions rest on the interpretation of the behavioral data:
1) The first behavioral test measuring the duration spent in the upper/ lower half of the arena was performed in groups of fish in the exposure arena. However, the diving response should be measured in a novel tank on individual fish. The authors do not address this issue while they clearly state that they have based this test on the novel tank diving test originally developed for zebrafish. The authors should first verify the use of this behavioral test, by showing that also brown trout display a diving response (not all fish species show this characteristic movement to the bottom!), and second, show that high concentrations of anxiolytics (SSRIs, benzodiazepines) affect the brown trout diving response. In the present form the sojourn preference results cannot be interpreted as anxiolytic. If these verifications can no longer be made because the experiment is finished, then these interpretation problems should at least be thoroughly discussed.
2) The reporting of the second behavioral test (ASMD) is insufficient to support the conclusion that control trout were startled while citalopram exposed trout were not startled. Ideally, velocity (in cm/s), velocity (in bodylengths/s) should be reported over time (in minutes) in addition to total duration not moving (in sec) over the whole trial. This to clearly show the initial startle response in the controls drove the observed treatment effect on velocity. At the moment, the lower velocity of the citalopram exposed fish could mean they were freezing more and citalopram therefore had an anxiogenic or sedative effect!

The first point can be solved by authors by discussing the differences between the sojourn preference and the novel tank diving test. The second point can be solved by adding more ethovision parameters and discussing possible alternative explanations for the findings in the ASMD.

Additional comments

- Line 28 remove "which could be suspected to alter behavioural and physiological parameters in fish". You could add this information to a separate sentence after this sentence, but at the moment this sentence is hard to read because of this subordinate clause.
- Line 29: with "most prevalent" do you mean that citalopram is the SSRI that is detected at the highest frequency or at the highest concentration?
- Line 32: please state exact concentrations and add between brackets which of these reflects the
environmentally relevant, low, medium and high concentration.
- Line 36 "sojourn": this is not a term that is often used in the fish behavior literature. I suggest changing it to the more commonly used "duration (of time) in" or "time spent in".
- Line 35 "decreased anxiety" is an interpretation. Anxiety is a human mental state and it is hard to investigate if fish experience this or how to measure this in fish. However you state explicitely which behavioral parameters measured which is very informative. Perhaps rephrase to "showed higher swimming activity and increased their time spent in the upper part of the arena, behavioral changes that are indicative of decreased anxiety".
- Line 39 "display": please change to "displayed"
- Line 51: "all surface waters" and "all trophic levels". This is not known since not every stream in the whole world has been sampled. There are probably still some pristine streams. Please change "all" to "most" or "most human-influenced waters".
- Line 53 "found in rising product types and concentrations in surface waters": it is unclear to me what you mean with "found in rising product types". Please consider rephrasing to "increasingly detected in rivers and surface waters". This more accurately reflects increases due to increased reporting/ more measurements as well as increased prescription rates.
- Line 53/ 54 reference to Fick et al. 2009: this paper is about pharmaceutical pollution by bulk production plants and therefore presents a special/ worst case. A more appropriate reference for surface water concentrations of psychoactive pharmaceuticals is Hughes et al. 2012 (global) or Loos et al. 2009 (EU), or Batt et al. 2016 (USA). Same for line 383.
Hughes, S.R., Kay, P., Brown, L.E., 2012. Global Synthesis and Critical Evaluation
of Pharmaceutical Data Sets Collected from River Systems. Environ. Sci.
Technol. 47, 661–677.
Loos, R., Gawlik, B.M., Locoro, G., Rimaviciute, E., Contini, S., Bidoglio, G.,
2009. EU-wide survey of polar organic persistent pollutants in European river waters. Environ. Pollut. 157, 561–568.
Batt, A.L., Kincaid, T.M., Kostich, M.S., Lazorchak, J.M., Olsen, A.R., 2016. Evaluating the extent of pharmaceuticals in surface waters of the United States using a National-scale Rivers and Streams Assessment survey. Environ. Toxicol. Chem. 35, 874–881.
- Line 57: change "(SSRI)" to "(SSRIs)"
- Line 58 "serotonin reuptake transporter": this transporter is called the serotonin transporter (SERT).
- Line 59: "presynapse membrane" please change to "presynaptic membrane"
- Line 68: please remove "in the effluent of drug manufacturers" and move the citation to Larsson et al. 2007 to the next sentence in order to separate the case of the outlet of bulk drug production sites from "ordinary" waste-water treatment plant effluents.
- Line 71/72: "one of the most common antidepressants" please rephrase to "one of the most commonly prescribed antidepressants".
- Line 62-71: the authors give a thorough overview of the citalopram concentrations detected in different environmental matrices. Since you make a comparison to fluoxetine in line 73, it would be interesting to compare the citalopram concentrations to fluoxetine concentrations (give approximate range in the environment). Also the potency and bioconcentration factor of both drugs should be taken into account when comparing these 2 SSRIs. Based on these considerations, Fick et al. 2010 have calculated that the predicted critical environmental effect concentration of citalopram is 141 ng L-1 while for fluoxetine is 489 ng L-1. A reference to this article could further strenghten your argument that citalopram is understudied.
Fick, Lindberg, R.H., Tysklind, M., Larsson, D.G.J., 2010. Predicted critical environmental concentrations for 500 pharmaceuticals. Regul. Toxicol. Pharmacol. 58, 516–523.
- Line 72: change "SSRI" to "SSRIs"
- Line 73: "like fluoxetine" : add a number of key references.
- Line 76: please add the species of stickleback used in this study. I think it is three-spined stickleback (Gasterosteus aculeatus).
- Line 75-90: indeed the results from different citalopram studies on fish have been quite contradictive even within the same species. I think this paragraph would benefit from placing the contractive results on the same species next to each other. For example, discuss Kellner 2015, 2016 and 2017 on stickleback in one go, highlighting that citalopram increased activity in the 2016 study while developmental exposure had the opposite effect on activity and even increased aggressive behavior (Kellner 2017). Another reason for the discrepancy between the stickleback studies could be the choice of behavioral test. The novel tank diving test was originally developed for zebrafish which show a characteristic movement to the bottom when entering a novel deep tank. In my experience, wild-caught stickleback do not show the same ”diving response” and hence results such as increased or decreased duration in the top should be interpreted with care.
- Line 100: what do you mean with ”sojourn preference”? This is not an established term.
- Line 126: how many replicate tanks were used per concentration and temperature?
- Line 128/129: 30 fish in 10L of water is a very high density that may affect water quality. Water changes twice a week may not be enough. Please provide data on ammonium levels. Same for experiment 2 (line 152-153).
- Line 137-138 ”At 7 days post-hatch, the heart rate of 5 individuals of each control and the highest concentration tank was measured and, whenever a difference was revealed, the other treatments were also assessed.” This strikes me as an unusual procedure to collect data. The choice of measurements to perform should be made independent of the outcome of the first measurements to avoid type I and type II errors.
- Line 138-140: It seems the duration spent in each vertical section of the aquarium was assessed in the exposure aquarium. First, it is unclear how you defined the vertical sections. But more importantly, measuring vertical movement in the home tank is quite a different thing from assessment of the diving response in the novel tank diving test, and might therefore give quite different results. In the novel tank diving test, the fish is placed alone in a novel, narrow and deep tank, a stressful situation for a shoaling fish species. Species such as zebrafish usually quickly move to the bottom, while this ”diving response” is reduced by chronic fluoxetine treatment (Egan et al. 2009 https://www.ncbi.nlm.nih.gov/pmc/articles/PMC2922906/ ) as well as treatment with the anxiolytic diazepam (Bencan 2009), therefore decreased duration at the bottom (or increased duration at the top) is usually interpreted as an anxiolytic effect. In your exposure tanks, there was likely a dominance hierarchy in the group which influenced where the fish swam, and there was no additional stressor of being alone in a novel tank.
- Line 146: why was the tailfin used for citalopram tissue concentration? Bioconcentration factors differ between different tissues and the most logical choice here would be brain tissue since this is the tissue where the drug exerts its effects. See Heynen et al (2016) for a comparison of pharmaceutical concentrations in different tissues of perch. https://www.publish.csiro.au/en/en16027
- Please provide a timeline to further clarify the sampling points and tests performed in each of the two experiments.
- Line 225: Please consider rephrasing ”artificial swimming measurement device (ASMD)” to ”automated video tracking of swimming behavior” or similar. Authors familiar with ethovision then immediately know what you mean, whereas ADSM is term that I have never encountered before. The use of the term ASMD also is unclear in line 141.
- Line 225-244: The ASMD procedure is unusual since multiple fish are tested in the same small test arenas (5 larvae in experiment 1 and 3 juveniles in experiment 2) instead of fish being tested individually. Also a key anxiety-related behavior, thigmotaxis (wall-hugging), has not been measured, which is a missed opportunity since this has been more closely linked to the behavioral effects of pharmacological effect concentrations of SSRIs compared to swimming activity. The small arenas may not have enabled the researchers to quantify thigmotaxis since this behavioral variable is best measured in larger test arenas that exceed the fish length by approximately 20 times.
- Line 241: the total distance moved should be directly proportional to average velocity if tracking is perfect after manual corrections of the track (i.e. if the total duration in the arena is exactly 18 minutes for each trial). Hence there is no need to report both variables. If tracking is not available for 100% of the trial, average velocity is more reliable since it compensates for the missing part of the track.
- Line 241: The authors calculated swimming velocity and total distance moved, but these parameters are influenced by the size of the fish. Larger fish swim larger distances. Indeed the fish exposed to the higher concentration are smaller (for both temperatures) and move shorter distances. It would be helpful to calculate swimming speed in bodylengths per second by dividing the velocity in cm/s by the measured standard length of each individual (in cm). Another ethovision parameter to add to table 1 and 2 is the duration of time spent moving/ not moving. This parameter sums the time intervals that the fish spent not moving (moving less than 2 cm/s say, this threshold should evaluated using integrated visualization as described in the ethovision manual) and is independent from swimming speed.
- Line 268: please indicate which response variables were transformed, and how.
- Line 271: it is not clear what is meant with ”block”.
- Line 385-411: ”bioaccumulation” is a term that is in some parts of the literature used to describe the accumulation over the food web. Please consider rephrasing to ”bioconcentration”. Please also consider adding a sentence in the beginning of this paragraph describing the bioconcentration factors for the different treatments used, for 2 reasons. First, it makes it more clear that for egg exposure, the tissue concentrations in fin muscle tissue are still lower than the water concentrations while in larvae exposure tissue concentrations exceed water concentrations. Also it makes it easier to compare bioconcentration factors for the different exposure concentrations. The calculation of the bioconcentration factor is found in Heynen et al. 2016 (https://www.publish.csiro.au/en/en16027 ).
- Line 395: ”possibly due to the longer exposure time of about 4 weeks” : please add the number of exposure days for each temperature between brackets.
- Line 401-406: could the authors please specify what the approximate citalopram concentrations were in the effluent studies (Grabicova et al. 2014, 2017)
- Line 413: change ”induced by citalopram” to ”influenced by citalopram exposure”
- Line 416: what is meant with ”exogenous feeding in salmonids is associated with higher mortality risks”? That predation risks are higher once the yolk sack is consumed? But the exposure tanks were covered in your experiment?
- Line 424: please provide exact fluoxetine concentrations in this study.
- Line 431: but developmental exposure to 1.5 ug L-1 citalopram increased food intake in stickleback (Kellner 2017). Can the authors please discuss that other studies have reported both increased and decreased food intake?
- Line 441-442: ”In general, the test design used in the present study for this parameter is comparable with the new tank diving test.” (Please change ”new” to ”novel”.) I very much disagree with this statement. There is a big difference between measuring the vertical position in the home tank in groups of fish, and measuring the diving response of individual fish in a novel tank. Zebrafish indeed show a clear 'diving response' where the fish initially lies still in the bottom in the first minutes followed by gradual movement to the middle and top zone. The authors first need to show that brown trout have the same response. The diving test has also been verified with acute and chronic exposure to therapeutic concentrations of anxiolytics and antidepressants, that reduce this behavior. This has also not been shown for brown trout. Hence the behavioral assay used here has not been verified for the species investigated and it is unclear how ”more fish in the top half” should be interpreted. These drawbacks and the differences with the NTDT should be made clear to the reader and cannot be ignored.
- Line 453-455: ”The significant difference in sojourn in the upper aquaria part of the brown trout larvae exposed to 1 and 10 μg/L citalopram at 7°C is more likely due to the fact that only one of the three replicate aquaria was analysed and therefore a single individual has a higher impact on the relative sojourn in the upper aquaria part.” Why was only 1 replicate tank analysed? This is in direct contrast to line 215 where you write ”Nevertheless, the selection of photographable tanks was representative for the entire number of aquaria.” Please provide information on the number of tanks observed for ”sojourn in upper half of the aquarium” for each treatment in table 1 and 2.
- Line 458: ”An anxiolytic effect of citalopram ...” please state exactly which behavioral variable was measured in these studies. Did these species also spent more time in the upper half of the arena?
- Line 461-462: Please move this sentence to the next section on behavioral activity. Also, you do not mention that Kellner et al. 2017 (stickleback developmental exposure, which is more comparable to the current study) also observed lower swimming activity, which is in line with your own results from the ASMD. Please provide a balanced review of the studies measuring behavioral activity in response to citalopram, highlighting any differences in results between studies and differences in methodology that could account for these divergent findings.
- Line 469: ”This effect is exclusively due to the anxiolytic and soothing effect of the antidepressant.” The relationship between anxiety-related behavior and behavioral activity is not at all as clear-cut as the authors propose here. In general, decreased activity is usually rather indicative of an anxiogenic and not an anxiolytic effect! Freezing is a very common response to a stressful situation in salmonids, and citalopram seems to increase this anxiogenic response! Hence you see an opposite effect of exposure to low concentrations of citalopram! This might have to do with the low concentrations used, and has been reported for low concentrations of fluoxetine as well. Fluoxetine exposed musquitofish (8 or 97 ng L-1) spent longer time freezing after the drop of a cilindrical metal probe into the tank (Martin et al., 2017). Wild-caught guppies exposed to fluoxetine (4 or 16 ng L-1 for 4 weeks) remained stationary for longer, spent longer time under plant cover and were less active following a simulated bird attack (Saaristo et al., 2017). An alterative explanation is that developmental exposure of larvae to citalopram can actually cause anxiogenic effects while anxiolytic effects are seen for adults. A third explanation is that the highest concentration of citalopram actually resulted in a sedative (sleep inducing) effect, which is thought to be caused by citalopram also being a mild antihistamine. Hence the current interpretation of reduced activity as anxiolytic is problematic and this part of the discussion lacks a critical review of the available literature.
Martin, J.M., Saaristo, M., Bertram, M.G., Lewis, P.J., Coggan, T.L., Clarke, B.O., Wong, B.B.M., 2017. The psychoactive pollutant fluoxetine compromises antipredator behaviour in fish. Environ. Pollut. 222, 592–599.
Saaristo, M., McLennan, A., Johnstone, C.P., Clarke, B.O., Wong, B.B.M., 2017. Impacts of the antidepressant fluoxetine on the anti-predator behaviours of wild guppies (Poecilia reticulata). Aquat. Toxicol. 183, 38–45.
- Line 471-474: ”Table 2: Results for juvenile brown trout exposed to citalopram. Data are shown as arithmetical mean ± standard deviation. Asterisks indicate significant differences to the respective controls (*p<0.05; **p<0.01; ***p<0.001). LoD=limit of detection” This information should be removed from the discussion as it is a figure caption.
- Line 476-477: ”which basically results in a startle reflex and increased escape behaviour”: no behavioral data is provided to show that the startle reflex drove the differences in activity between citalopram treatments. Clear evidence would be increased velocity in the beginning of the trial which gradually is reduced over the 20 minutes of the trial, with citalopram exposed fish showing less of a peak in velocity in the beginning. Considering the rather low average velocities measured in this experiment (2 cm/s and 1 cm/s or lower in the high concentration) I would rather presume that the differences in activity were driven by differences in time spent freezing.
- Line 478-479: ”as they swam slower and less bustling than control fish” Again, no data is provided to show that control fish were burst swimming (not bustling) more. As discussed earlier, the movement (duration moving/ not moving) variable from ethovision might back this up, but velocity/ distance moved by itself does not give any information on whether the control fish moved with higher routine swimming speed or alternatively, whether their speed was higher because they were bursting more. Please provide evidence!
- Table 1 and 2: what do the standard deviations represent? Variation between replicate tanks? Variation between individuals?
- Figure 1: Please consider dividing up this graph into 8 bars: 4 citalopram concentrations that were directly dissected for cortisol and 4 citalopram concentrations tested in the ASMD. This will make your general point more clear that citalopram exposure did not affect cortisol concentrations, while the ASMD method was stressful to the fish.
- Please add a graph with the ASMD data: velocity in bodylengths per second over time (in minutes) per citalopram concentration.

---

## Round 0.2 · Minor Revisions

There are a few more reviewer suggestions that I would like you to address. I believe these will improve your manuscript even further. Thanks for your efforts to date in improving your work in response to previous reviewer comments.

Reviewer 1 ·

Basic reporting

Line 89: “1,5 ug/L” — should be 1.5

Line 91: “1,5 ug/L” — should be 1.5

Line 96: “2,3 and 15 μg/L” – should be 2.3

Line 85–102: It’s good to see an open discussion of the different results across studies. However, when listing the results of the different experiments it’s difficult to ascertain when results are in accordance or contrasting. Could the authors please make this clearer. Study X found a reduction in anxiety-related behaviour at X ug/L with a X day exposure, whereas Study X reported no change in anxiety at X ug/L with a X day exposure. The comparisons of studies addressing feeding were done well.

Line 140: “static three block setup to 0, 1, 10, 100, 1000 μg/L citalopram at both 7°C and 11°C” — to avoid confusion should read ‘at 7°C or 11°C’. Also, it should read three-block'

Line 244: “Small aquaria (17*17*8.5 cm)” — typically formatted with ‘×’ not ‘*’

Line 249–250: “(Basler acA 1300-60 gm, 1.3 MP resolution, Basler AG, Ahrensburg, Germany, lens: 4.5-12.5 mm; 1:1.2; IR 1/2”)” — When giving a range of number you should use N-dash (–), not a hyphen (-). For example, “1300-60 gm” should be ‘1300–60 gm”.

Line 251: “The setup” — ‘set-up’

Line 275: “3,3’,5,5” – It appears that you have used commas, as decimal points throughout. I would recommend changing this.

It is uncommon to report p-values beyond 3 decimal places. It will help readability if you reduce to 3 decimal places.

Line 332–333: “During this exposure time, fish were 43 d – 71 d post hatch which is the critical phase of complete yolk sac consumption and first food intake.” – This information belongs in the discussion or intro, not results.

You may find a recent paper on reporting disputed foraging dynamics of fish exposed to the SSRI fluoxetine useful here, given your fish also exposed in groups.
Martin, J. M.; Saaristo, M.; Tan, H.; Bertram, M. G.; Nagarajan-Radha, V.; Dowling, D. K.; Wong, B. B. M., Field-realistic antidepressant exposure disrupts group foraging dynamics in mosquitofish. Biology Letters 2019, 15, 20190615.

Experimental design

Swimming behaviour in exposure aquaria: the data gathered with this experiment is a little hard establish, as it is currently worded. Is the following interpretation correct? If so I would suggest re-wording as follows…

‘During the exposure, over a 10-day period, three pictures were taken daily across X tanks. For each day, an average was calculated using the three photographs, giving a total of 10 daily measures for analysis (n = X per treatment).’

I used ‘X’ tanks because I wasn’t actually sure how many tanks you measured. Please make this clear.

Line 231–232: “Nevertheless, the selection of photographable tanks was representative for the entire number of aquaria” — It is unclear what is meant by this sentence.

Validity of the findings

You need to report how much of the variation was captured by your random effects.

Additional comments

It was my pleasure to re-review this article for PeerJ. I thank the authors for taking on board previously feedback. I have made some additional suggestions to improve clarity, particularly in the swimming behaviour assay. I have also requested the addition of some statistical information, namely the variability captured by the random effects. I believe that with the following changes the MS will be suitable for publication. Excellent work.

Reviewer 2 ·

Basic reporting

no comment

Experimental design

no comment

Validity of the findings

Overall the conclusions are thoroughly supported by the data. Only the newly added data on velocity and no movement over time lacks statistical analysis (see general comments).

Additional comments

In my opinion, the manuscript has been greatly improved since the first version. Especially the interpretation of the behavioral results (vertical distribution of fish during exposure and the ASMD) has been expanded and there is now also a stronger link to previous studies. I only have a couple of minor comments:
- Line 72: change “range” to “ranging”.
- Line 86: add “part of the” between the words “upper” and “aquaria”.
- Line 139: “three block” is not an established term. Please consider rewriting to “a semi static setup with three replicates each of 0, 1, 10, 100, 1000 µg/L citalopram at both 7°C and 11°C”
- Line 368-373: please provide statistical proof for these statements. In figure 1, panel B, C, E, F, H and I it is also difficult to compare the control groups with the exposed groups. Please consider converting these bar graphs to line bars with a single line per treatment connecting dots on the lines which indicate the mean per minute per treatment. The same comment holds for lines 391-394, please provide statistical proof here as well.
- Line 473-487: the discussion of the interpretation of the swimming behavior during exposure has been greatly improved. The comparison with both the scototaxis and novel tank diving test as well as shoaling behavior has been expanded which greatly facilitates the interpretation of the behavioral results (vertical distribution) obtained from groups during exposure.
- Line 495: please consider changing “quantity spent” to “time spent”
- Line 508: please consider rephrasing “except of decreased acitivty” to “except for decreased activity”.
- Line 529-536: also the discussion of the ASMD has been improved and I agree with your take-home message of a sedative effect.
- Line 540-544 is a repetition of line 493. Perhaps simply refer to this earlier piece: “Similar to the effect on citalopram on vertical distribution patterns during exposure, behavioral changes in the ASMD were more pronounced in exposed larvae than in juveniles. As previously mentioned this might be the result of the longer exposure of larvae or differences between life stages in sensitivity (Schwarz et al. 2017).”
- Line 586-7: Please consider rephrasing “which cannot exclusively be associated with the longer exposure time, but also different sensitivities can play a role” to “which could be associated with the longer exposure time in larvae compared to juveniles, but also differences in sensitivity between life stages may play a role”

---

## Round 0.3 · accepted · Accept

Thanks for your patience, the article is now Accepted.